# BIG-LAYERS: ENABLING END-TO-END TRAINING

## ABSTRACT

Training deep neural networks on extremely large inputs—such as gigapixel Whole Slide Images (WSIs) in digital pathology—poses significant challenges due to GPU memory constraints. Multiple Instance Learning (MIL) circumvents this limitation by processing patches from a WSI. However, the encoder used to get patch embeddings is usually a generic pre-trained deep neural network model. In this paper, we propose a training strategy that enables training the encoder by dynamically offloading intermediate activations of a layer to CPU RAM, allowing the layer to process inputs that do not fit in the GPU memory. We demonstrate the effectiveness of our approach on PANDA and CAMELYON datasets using popular MIL approaches. Experimental results indicate that our method improves the Quadratic Weighted Kappa (QWK) metric, on PANDA, by 7–15 percentage points compared to ResNet-18 baselines where encoders are kept frozen. Evaluations on external test sets further suggest better generalisation, and in some configurations, our models even outperform foundation-model encoders on TCGA-PRAD. The code will be made publicly available upon publication.

## 1   INTRODUCTION

Advances in deep learning have revolutionised histopathology (Unger & Kather, 2024; Van der Laak et al., 2021), but some challenges in handling Whole Slide Images (WSIs) remain. One of the challenges is the enormous size of WSIs, which can be up to a few gigapixels. It prevents the use of common machine learning techniques, as these techniques require much smaller images to be directly applicable. For classification tasks, a common approach to handle such large images is to use Multiple Instance Learning (MIL) (Dietterich et al., 1997; Maron & Lozano-Pérez, 1997) in which some patches are extracted from the WSI, and it is assumed that a subset of those patches corresponds to the desired label. It is also sometimes referred to as weakly supervised learning. MIL involves three steps: first, an encoder, such as a Convolutional Neural Network (CNN), converts a patch into an embedding; second, an aggregator pools the embeddings into an aggregated embedding; and third, a classifier assigns a label to the aggregated embedding.

However, due to the large number of patches required to train a model effectively, training a model end-to-end is practically infeasible on most GPUs. Therefore, the common approach is to use a pre-trained encoder to extract the embeddings for all the patches and train only the aggregator and the classifier models (Song et al., 2023).

Foundation models—large, heavily pre-trained encoders learned from massive histopathology image corpora—have become increasingly popular, prompting a rapid proliferation of models developed for a wide range of computational pathology tasks. Despite their rapid adoption, accumulating evidence shows these models can be brittle under realistic distribution shifts (de Jong et al., 2025). Recent evaluations show that pathology foundation models often encode non-biological technical signals—such as medical-center, scanner and staining signatures—that undermine their robustness under distribution shift (Kömen et al., 2024; de Jong et al., 2025; Gustafsson & Rantalainen, 2024). Linear-probe analyses further reveal that these site- and batch-specific factors are readily recoverable from foundation-model embeddings and can dominate biologically relevant variation, leading to systematic errors when models are deployed across institutions (Kömen et al., 2024; de Jong et al., 2025; Kömen et al., 2025). Moreover, standard mitigation strategies, including stain normalisation, distillation and larger-scale pretraining, only partially reduce these vulnerabilities (Kömen et al., 2024; 2025; Gustafsson & Rantalainen, 2024; Filiot et al., 2025). These findings underscore the need

for rigorous robustness assessment and task-aware adaptation when applying foundation encoders in heterogeneous clinical environments.

Recent work has attacked the GPU-memory bottleneck for whole-slide and other multi-megapixel inputs using several complementary strategies. Some authors leverage CUDA unified memory to let the runtime page very large tensors between host and device (Chen et al., 2021); others stream or tile the input so that convolutional layers run on spatial tiles and intermediate outputs are stitched (Pinckaers et al., 2020); still others reduce GPU state by offloading optimizer/parameter state to CPU (Ren et al., 2021) or running the backward pass for only a subset of patches (Skrede et al., 2020).

In this work, we introduce a training strategy that allows comprehensive end-to-end training of models by efficiently using CPU RAM and GPU RAM for layers whose input and output are too big to fit in the GPU memory, including layers that require computing statistics over the entire input. We demonstrate the utility of our approach by training models on the PANDA dataset (Bulten et al., 2022) using ResNet18 (He et al., 2016) as the encoder. Models trained using our method perform significantly better on the test set compared to baseline models where the encoder is frozen, gaining several percentage points on Cohen's quadratic weighted Kappa $\kappa^2$ (QWK) metric.

## 2 RELATED WORK

Memory- and I/O-aware techniques for training on large inputs have followed several broad paradigms; we summarise each and explain how our method differs.

**Unified / runtime-managed memory**    Chen et al. (2021) demonstrate that CUDA unified memory can enable end-to-end training on entire whole-slide images by relying on the CUDA runtime to page tensors between host and device. This approach simplifies implementation because the runtime performs paging implicitly, but it offers limited control over transfer scheduling. In practice, the explicit high-level unified-memory knobs used in older TensorFlow releases are not exposed as stable TensorFlow version 2 public APIs. Unified memory may also be suboptimal compared to methods that explicitly manage data transfers between GPU and CPU (Landaverde et al., 2014; Jarząbek & Czarnul, 2017; Alawneh et al., 2018). Our method does not depend on CUDA unified memory; instead, we perform layer-aware transfers and precisely control when and how tensors move between CPU and GPU.

**Streaming / tiled convolutional training**    Pinckaers et al. (2020) split the input into spatial tiles and execute convolutions tile-by-tile, stitching intermediate feature maps. They use this idea to train ResNet architectures end-to-end on large Whole Slide Images. They use gradient checkpointing (Chen et al., 2016) to avoid storing intermediate representations for those layers. While this alleviates memory constraints, it can not be used to train layers that require computing statistics over the entire input, which includes common layers such as batch normalisation. By contrast, we present a more generic method for implementing commonly used layers. Our method enables training layers that require global statistics.

**Partial backward / selective gradient updates**    Skrede et al. (2020) reduce memory consumption by computing gradients for the encoder only for selected patches. Their approach reduces memory and computation at the cost of computing approximate gradients; they are effective when full gradients are unnecessary but can harm representational learning when end-to-end gradient fidelity matters. In contrast, we compute gradients for all the tiles processed by the encoder.

**Optimizer/state offload (ZeRO-offload)**    Ren et al. (2021) design ZeRO-Offload to reduce GPU memory pressure by moving model state and optimizer work onto CPU. ZeRO-Offload partitions model states and keeps model parameters on the GPU while offloading averaged gradients, and the optimizer update computation to the CPU. The approach uses a highly optimized CPU Adam (Kingma, 2014) implementation and enables training large models on a single GPU. By contrast, our method targets memory arising from extremely large single-example tensors rather than the whole model state or the optimizer. Our approach enables training layers whose activation does not fit in the GPU. ZeRO-Offload and our approach are complementary and can be combined.

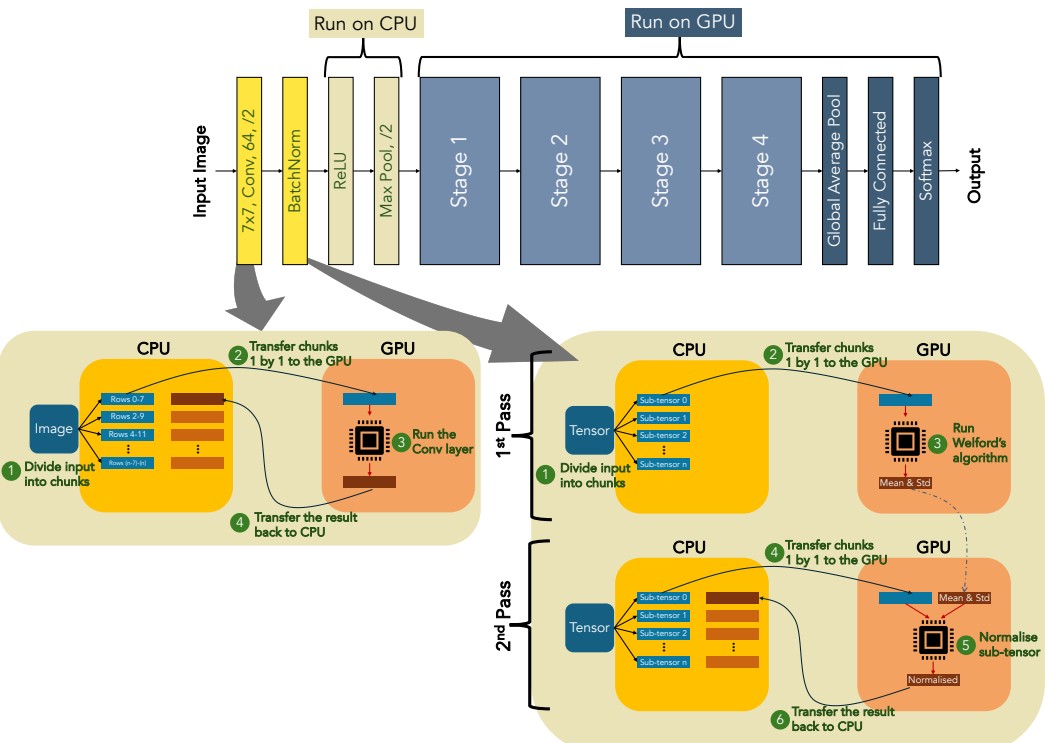

Figure 1: ResNet18 architecture implemented using our method.

# 3 PROPOSED METHOD

## 3.1 OVERVIEW

Our method allows training neural networks when the input and output of one or more layers do not fit in the GPU memory. To achieve that, our method employs the following key strategies:

- **Partitioning:** Large tensors are divided into sub-tensors that fit within GPU memory.

- **Selective Offloading:** Intermediate activations that would otherwise exceed GPU capacity are stored in CPU RAM and transferred back to the GPU only when needed.

- **Layer-Specific Execution:** Compute-intensive layers (e.g., convolutional layers and batch normalisation layers) leverage the partitioning and selective offloading strategies on the GPU, while computationally cheap layers (e.g., activation layers such as ReLU and pooling layers such as MaxPool) are executed on the CPU.

- **Efficient Backpropagation:** The same data partitioning and offloading techniques are applied to gradient computations, ensuring a memory-efficient backward pass.

Typically, the first few layers of modern CNN architectures progressively downsample the input image. These layers consume a lot of memory, but as the network deepens, the feature maps become smaller, and the memory requirements decrease substantially. Our approach utilises the CPU RAM to process the initial high-memory-demand layers. Once sufficient downsampling has occurred, the data remains on the GPU for the rest of the network. How many layers should leverage the CPU RAM can easily be adapted to the particular GPU setup using our implementation.

Algorithm 1 outlines our method. Figure 1 illustrates our implementation of ResNet18 based on Algorithm 1.

---

**Algorithm 1** Our method

---

**Require:** Image $X$, layers $\{L_i\}_{i=1}^N$ of the neural network
**Ensure:** Output tensor $Y$
 1: **procedure** EFFICIENTFORWARD($X, \{L_i\}_{i=1}^N$)
 2:     **for** each layer $L_i$ in the network **do**
 3:         **if** $L_i$ is compute-intensive (e.g., Conv, BatchNorm) **then**
 4:             **Partition** $X$ into sub-tensors $\{X_j\}$ that fit in GPU memory
 5:             **for** each sub-tensor $X_j$ **do**
 6:                 Transfer $X_j$ to GPU
 7:                 Compute $Y_j \leftarrow L_i(X_j)$ on GPU
 8:                 Transfer $Y_j$ back to CPU if subsequent layers require partitioning
 9:             **end for**
10:         **else if** $L_i$ is computationally inexpensive (e.g., ReLU, MaxPool) **then**
11:             Compute $Y \leftarrow L_i(X)$ directly on CPU
12:         **end if**
13:         $X \leftarrow Y$                ▷ Update input for the next layer
14:         **if** tensor size has been significantly reduced **then**
15:             Transfer entire $X$ to GPU for remaining layers
16:         **end if**
17:     **end for**
18:     $Y \leftarrow X$                   ▷ Store final output
19:     **return** $Y$
20: **end procedure**

---

## 3.2 Partitioning and selective offloading

The core of our method for handling computationally heavy layers is to partition the input tensor into sub-tensors and incrementally compute the output using those sub-tensors. First, we divide the input tensor into (potentially overlapping) sub-tensors. Then, we transfer sub-tensors to the GPU one by one and perform the layer-specific computation with the sub-tensor present on the GPU. Some layers, like normalisation layers, require repeating the previous step to get the final output. For example, for BatchNorm, we first compute the mean and standard deviation of the input using Welford's algorithm in the first sequential transfer of sub-tensors, followed by a second transfer to normalise the input using the mean and standard deviation computed in the first pass. For the convolution layer, we transfer the sub-tensors only once, but the sub-tensors might have an overlap depending on the stride used in the layer.

Since the concrete implementation of our generic method differs from layer to layer, we detail the implementation of the forward pass for BatchNorm in Algorithm 2 and illustrate it in Figure 1, demonstrating how to use the generic method for a layer that requires global statistics. We provide a similar implementation of the Convolution layer in Appendix A.3.

## 3.3 Limitations

Our methodology involves partitioning the input tensor into sub-tensors and executing computations incrementally using these sub-tensors. Although all commonly used layers can be computed in this manner, any computationally intensive layer whose computation can not be decomposed in this way will present a significant constraint as it will necessitate computation on the CPU, which can be prohibitively expensive.

Architectures that do not significantly downsample the input in the early layers or do not downsample at all (like Vision Transformers (ViTs) (Dosovitskiy, 2020), which do not downsample except for the initial embedding layer) require using our method for most (or all) layers in the network. This can be too slow to be practically useful, especially for very big networks; however, self-attention can be implemented efficiently using tiled, IO-aware kernels that align with our partition-and-offload strategy.

---

**Algorithm 2** Memory-efficient Batch Normalisation (Forward Pass)

---

**Require:** Input tensor $X$, scale parameter $\gamma$, shift parameter $\beta$, running mean $r\_mean$, running variance $r\_var$, maximum count $max\_N$ of tensor elements to transfer to GPU, small constant $\epsilon$

**Ensure:** Normalized output tensor $Y$, updated running statistics $(r\_mean, r\_var)$

 1: **procedure** BIGBATCHNORMFORWARD($X$, $\gamma$, $\beta$, r_mean, r_var, max_N, $\epsilon$)
 2:     **Partition** $X$ into sub-tensors $\{X_i\}$, each with at most max_N elements
 3:     **if** training **then**
 4:         **for** each sub-tensor $X_i$ of $X$ **do**
 5:             Transfer $X_i$ to GPU
 6:             Update variables in Welford's algorithm
 7:             Transfer $Y_i$ back to CPU memory
 8:         **end for**
 9:         **Update** running statistics r_mean and r_var
10:     **else**
11:         Set $\mu \leftarrow$ r_mean and $\sigma^2 \leftarrow$ r_var
12:     **end if**
13:     **Normalize:**
14:     **for** each sub-tensor $X_i$ of $X$ **do**
15:         Transfer $X_i$ to GPU
16:         Compute:

$$Y_i \leftarrow \gamma \cdot \frac{X_i - \mu}{\sqrt{\sigma^2 + \epsilon}} + \beta$$

17:         Transfer $Y_i$ back to CPU memory
18:     **end for**
19:     **return** $Y$, r_mean, r_var
20: **end procedure**

---

Implementing self-attention with our method is straightforward in principle. Self-attention reduces to a sequence of linear projections and small matrix multiplications, and linear layers map directly to the same partitioning approach we apply to convolutional layers (a 1×1 convolution is algebraically equivalent to a linear layer on flattened inputs). In practice, algorithms like FlashAttention (Dao et al., 2022) split the attention computation into small blocks, process each block on the GPU, and update outputs incrementally. This block-wise approach fits naturally with our sub-tensor transfer strategy, enabling memory-efficient self-attention.

## 4 EXPERIMENTS

### 4.1 DATASETS

**PANDA** This dataset (Bulten et al., 2022) consists of 10616 WSIs of hematoxylin and eosin (H&E)-stained needle biopsy WSIs of prostate tissue from two medical centres. Each WSI carries an International Society of Urological Pathology (ISUP) grade - 0 for normal tissue and 1 to 5 for cancer, forming a 6-class classification task. All slides are in 20× resolution.

**TCGA-PRAD** This dataset contains 449 resection WSIs from The Cancer Genome Atlas (TCGA) repository of prostate adenocarcinoma Zuley et al. (2016). After a pathologist's review, we removed 5 WSIs lacking tumour and 23 that could not be opened. The remaining 421 WSIs represent 394 unique patients; we randomly selected one WSI per patient to use as an external test set.

**CAMELYON17** This dataset (Litjens et al., 2018) comprises 1000 H&E-stained WSIs from five Dutch medical centres (with five slides per patient in the released patient-centric setup) and—when aggregated with Camelyon16—forms a collection of 1399 annotated WSIs. CAMELYON17 provides patient-level pN-stage labels (aggregating slide-level findings) and includes a subset of lesion-level manual annotations (10 annotated slides per centre in the training set) to support both slide-level classification and lesion localization tasks. All slides are in 40× resolution.

**CAMELYON16** This dataset (Bejnordi et al., 2017) contains 399 WSIs of H&E-stained sentinel lymph node sections collected from two Dutch centres. The dataset splits into 270 training slides and 129 test slides; the training slides include pixel-level delineations of metastatic regions provided as XML contours and binary masks. All slides are in $40\times$ resolution.

## 4.2 DATASET PREPARATION

We tile the WSIs into non-overlapping $256\times256$ patches. For PANDA and TCGA-PRAD, we retain only the patches with at least 60% foreground pixels. We convert each tile to greyscale and consider pixels with intensities between 3 and 230 as foreground. For CAMELYON 16/17, we follow Zhang et al. (2022) for tiling the WSIs, tiling them at $20\times$ resolution after localising the tissue region using OTSU's threshold method (Otsu et al., 1975).

### 4.2.1 PANDA SPLITS

We use the training/validation/test split from Song et al. Song et al. (2024), which provides a label-stratified division of 80:10:10 after removing 1061 noisy WSIs, resulting in 7647, 954, and 954 WSIs for the training, validation, and test subsets, respectively. We train all models exclusively on the PANDA training subset and evaluate them on its test subset. Additionally, we use TCGA-PRAD as an external test set to further assess generalisability.

### 4.2.2 CAMELYON SPLITS

We train exclusively on the CAMELYON17 training set and follow the reprocessed binary labels proposed by Ling et al. (2025). From CAMELYON17's training set we randomly select 50 WSIs to form a validation set and use the remaining 472 WSIs for training. We evaluate on the official CAMELYON17 test set and use CAMELYON16 as an external test set.

### 4.2.3 METHODS

We evaluate three methods: Attention-based MIL (ABMIL) Ilse et al. (2018), Double-tier feature distillation MIL (DTFD) Zhang et al. (2022), and TransMIL Shao et al. (2021). For each method, we compare baseline models that freeze the ResNet-18 encoder with our approach that trains it. All models are initialised with a ResNet-18 encoder pre-trained on ImageNet Deng et al. (2009), and we use its final stage output as the patch embedding. In addition, we evaluate three publicly released foundation encoders: H-optimus-1 (Bioptimus, 2025), UNI2-h (released alongside UNI) (Chen et al., 2024), and Prov-GigaPath (Xu et al., 2024). For foundation encoders, we follow the baseline hyperparameter settings applying the same optimisation schedule and learning-rate choices as used for the baseline frozen-ResNet models. We train three models per method and select the best checkpoint based on the validation QWK score.

### 4.2.4 TILE SELECTION

During training, we randomly sample 256 tiles per WSI to form a bag. During testing, we use all foreground tiles; for TCGA-PRAD, we additionally evaluate on 256 randomly selected tiles per WSI. As the specific 256 tiles may vary across random seeds, we run 100 tests per model using the mean QWK as the final QWK for the model.

For the CAMELYON experiments, we adopt method-specific sampling strategies that work best for the method. For baseline models we construct training bags by randomly sampling 1024 tiles per WSI. For our models we sample 512 tiles per WSI and use a batch size of 2. During evaluation, for our DTFD and TransMIL models we sample 2048 tiles per WSI, repeat the evaluation 3 times, and use the mean of those 3 runs as the final score for a given model. For our ABMIL models and all baseline models, we use all foreground tiles at test time.

### 4.2.5 OPTIMISATION HYPERPARAMETER SETTINGS

**PANDA** We test two hyperparameter configurations per method. In the first, we select a learning rate from $1e{-}4, 5e{-}5, 1e{-}5$ and train for 20 epochs with cosine annealing (Loshchilov & Hutter, 2016), a batch size of 2, and gradient accumulation over 16 steps, inspired by Song et al. (2024).

Table 1: Cohen's quadratic weighted kappa (QWK) on the PANDA dataset's test subset and TCGA-PRAD. We train three models per method and report the mean and standard deviation of QWK. For TCGA-PRAD, we also report QWK using only 256 randomly selected tiles per scan. We test each model 100 times when using only 256 tiles and use the mean QWK of the 100 runs as the final QWK for that model. "-Aug" indicates the use of augmentation.

| Method | | PANDA | TCGA-PRAD-All | TCGA-PRAD-256 |
|---|---|---|---|---|
| H-optimus-1 | ABMIL | $94.04 \pm 0.30$ | $68.11 \pm 2.21$ | $66.60 \pm 1.47$ |
| | DTFD | $93.18 \pm 0.44$ | $59.28 \pm 1.77$ | $66.34 \pm 0.49$ |
| | TransMIL | $95.56 \pm 0.39$ | $46.19 \pm 10.48$ | $54.10 \pm 4.56$ |
| Prov-GigaPath | ABMIL | $93.91 \pm 0.25$ | $67.53 \pm 1.10$ | $64.71 \pm 1.28$ |
| | DTFD | $92.90 \pm 0.29$ | $61.94 \pm 1.31$ | $64.10 \pm 0.42$ |
| | TransMIL | $94.14 \pm 0.43$ | $52.30 \pm 5.19$ | $53.89 \pm 2.89$ |
| UNI2-h | ABMIL | $93.61 \pm 0.12$ | $71.10 \pm 0.72$ | $68.65 \pm 0.63$ |
| | DTFD | $93.20 \pm 0.15$ | $69.98 \pm 0.73$ | $67.55 \pm 1.04$ |
| | TransMIL | $94.35 \pm 0.38$ | $55.62 \pm 5.98$ | $58.43 \pm 4.27$ |
| ABMIL | Baseline | $76.74 \pm 0.43$ | $56.34 \pm 0.85$ | $47.31 \pm 1.18$ |
| | + Ours | $84.60 \pm 1.15$ | $53.53 \pm 2.35$ | $49.61 \pm 1.62$ |
| DTFD | Baseline | $73.64 \pm 1.48$ | $55.26 \pm 1.87$ | $47.83 \pm 0.74$ |
| | + Ours | $87.13 \pm 0.63$ | $64.54 \pm 2.97$ | $56.04 \pm 2.42$ |
| TransMIL | Baseline | $81.87 \pm 1.13$ | $44.63 \pm 3.16$ | $46.86 \pm 1.70$ |
| | + Ours | $89.60 \pm 0.52$ | $29.96 \pm 9.12$ | $52.49 \pm 1.67$ |
| ABMIL-Aug | Baseline | $77.89 \pm 0.50$ | $52.83 \pm 0.87$ | $51.29 \pm 0.69$ |
| | + Ours | $86.91 \pm 1.31$ | $65.86 \pm 1.73$ | $63.46 \pm 0.89$ |
| DTFD-Aug | Baseline | $71.53 \pm 1.18$ | $56.79 \pm 4.21$ | $49.21 \pm 2.44$ |
| | + Ours | $86.59 \pm 0.40$ | $69.47 \pm 0.76$ | $62.65 \pm 0.54$ |
| TransMIL-Aug | Baseline | $78.06 \pm 1.06$ | $36.25 \pm 0.78$ | $44.69 \pm 2.54$ |
| | + Ours | $89.39 \pm 0.85$ | $45.56 \pm 7.15$ | $65.33 \pm 1.14$ |

In the second, we train for 45 epochs without cosine annealing or gradient accumulation, applying exponential decay (rate 0.955). All experiments use the Adam optimiser (Kingma, 2014) with a weight decay of $1e-4$.

**CAMELYON17** For baseline models we train with a learning rate of $1e-4$ for 120 epochs. For our models we train with a learning rate of $5e-5$ for 90 epochs. Both baseline and our models use a cosine-annealing learning-rate schedule and gradient accumulation over 2 steps. Baseline models use Adam optimiser while our models use Adam optimiser with DEMON momentum decay rule (Chen et al., 2022). Adam is used with weight decay of $1e-4$.

### 4.2.6 INPUT AUGMENTATION

We train models with and without augmentation. We employ Gaussian blur, colour jitter, random horizontal and vertical flips, and random rotation. We apply a single set of randomly selected parameters uniformly per WSI rather than augmenting each tile independently.

## 5 RESULTS AND DISCUSSION

### 5.1 PANDA AND TCGA-PRAD

Table 1 shows that end-to-end training with our method improves the QWK by 7 to 15 percentage points on the PANDA test set. On TCGA-PRAD, the baseline outperforms ABMIL and TransMIL without augmentation; however, our method with augmentation attains superior results. Baseline

models that freeze the encoder do not benefit from augmentation, whereas our approach exploits it effectively.

TransMIL exhibits high variance when tested on all tiles. We hypothesise that these inferior results stem from a training-testing mismatch. Specifically, all models train on bags with 256 tiles per WSI. While the PANDA test set averages 400 tiles per WSI, with the maximum being 1400 tiles, TCGA-PRAD averages 12800 tiles with the maximum being 41000. Because TransMIL employs self-attention that directly processes inter-tile interactions, it likely performs suboptimally on WSIs with approximately 50 times more tiles than WSIs seen during training.

We validate this hypothesis by testing on TCGA-PRAD using a subset of 256 randomly selected tiles per WSI. To handle variability from random selection of tiles, we run the evaluation 100 times and use the mean QWK as the final QWK for each model. Under these conditions, TransMIL improves notably, reaching QWK values comparable to other methods. Moreover, our approach with augmentation also maintains superior generalisability on TCGA-PRAD.

Foundation models consistently yield stronger in-domain performance on the PANDA test set, but the picture reverses on the external cohort when we compare each method using its best TCGA-PRAD result (taking the better of TCGA-PRAD-All and TCGA-PRAD-256 per method/approach). Under this best-of-two comparison, foundation models remain superior for ABMIL; for DTFD only the UNI2-h encoder slightly outperforms our augmented variant by 0.5%; and for TransMIL our augmented TransMIL (evaluated with the 256-tile protocol) achieves the highest external QWK overall, exceeding all three foundation-encoder variants. These results reinforce recent reports that foundation features often deliver excellent in-domain accuracy but can falter on out-of-distribution cohorts.

## 5.2 CAMELYON

Table 2 reports accuracy on the CAMELYON17 test subset and the CAMELYON16 whole set. End-to-end training with our method increases accuracy by up to 6 percentage points. Our models generalise more effectively, demonstrating larger performance improvements on the external CAMELYON16 test set compared to the internal CAMELYON17 test subset.

Many whole-slide images in both CAMELYON datasets contain substantially more tiles—up to an $80\times$ increase—than the 512-tile bags we use for training; accordingly, we observe high evaluation variance for both DTFD and TransMIL when we evaluate on all tiles. To address this training–testing mismatch, we evaluate every model on two setups: (1) all foreground tiles per WSI and (2) 2048 randomly sampled tiles per WSI (the same number we use in the validation set for checkpoint selection). We repeat the 2048-tile evaluation 10 times and use the mean accuracy as the final score for a given model. On the CAMELYON17 test set, our models generally achieve higher accuracy in the 2048-tile evaluation setup, whereas the baseline models show comparable performance across both setups. In contrast, on CAMELYON16, most methods perform better when evaluated on all tiles, with the sole exception of our TransMIL models. This pattern is consistent with the PANDA results, where TransMIL exhibits improved performance on external datasets when tested with a limited number of tiles.

Whereas foundation models showed mixed external behaviour in the PANDA–TCGA analysis, here they offer clear advantages on both datasets: they match or exceed our models on the internal CAMELYON17 test set and deliver consistently stronger performance on CAMELYON16. This could indicate that the domain gap between CAMELYON17 and CAMELYON16 is smaller than between PANDA and TCGA-PRAD.

We observe similar trends for the AUC metrics, reported in Table 3 in Appendix A.1.

## 5.3 SPEED

**Training epoch comparison.** We report per-epoch wall-clock time for three setups on PANDA (NVIDIA RTX 3090, 24 GB): precomputed-embeddings (train only MIL head) takes $\approx 20$ s/epoch; baseline on-the-fly (compute embeddings each iteration, no encoder backpropagation) takes $\approx 45$ min/epoch; end-to-end (encoder backpropagation using our approach) takes $\approx 200$ min/epoch.

Table 2: Accuracy on the CAMELYON17 dataset's test set and CAMELYON16 whole dataset. We train three models per method and report the mean and standard deviation of accuracy. We report accuracies using 2 different inference setups: 1) Using all foreground tiles of the WSIs 2) Using only up to 2048 radomly selected tiles per WSI. For the 2048-tiles per bag setup, we test each model 10 times and use the mean accuracy of the 10 runs as the final accuracy for that model. "-Aug" indicates the use of augmentation.

| Method | | CAM17-All | CAM17-2048 | CAM16-All | CAM16-2048 |
|---|---|---|---|---|---|
| H-optimus-1 | ABMIL | $88.64 \pm 0.92$ | $90.52 \pm 0.40$ | $96.84 \pm 0.34$ | $95.15 \pm 0.20$ |
| | DTFD | $87.75 \pm 1.45$ | $90.26 \pm 0.72$ | $94.35 \pm 2.00$ | $93.98 \pm 0.65$ |
| | TransMIL | $87.12 \pm 1.49$ | $89.11 \pm 0.73$ | $93.16 \pm 4.06$ | $92.52 \pm 3.23$ |
| Prov-GigaPath | ABMIL | $89.92 \pm 0.69$ | $91.43 \pm 0.27$ | $96.48 \pm 1.42$ | $94.21 \pm 0.48$ |
| | DTFD | $87.63 \pm 0.94$ | $90.76 \pm 0.46$ | $94.09 \pm 0.84$ | $92.31 \pm 0.45$ |
| | TransMIL | $89.36 \pm 1.32$ | $91.01 \pm 0.93$ | $95.28 \pm 1.12$ | $93.65 \pm 0.66$ |
| UNI2-h | ABMIL | $87.63 \pm 0.90$ | $89.60 \pm 0.54$ | $93.11 \pm 2.10$ | $93.29 \pm 1.16$ |
| | DTFD | $88.81 \pm 0.56$ | $90.83 \pm 0.40$ | $93.78 \pm 3.16$ | $94.22 \pm 1.44$ |
| | TransMIL | $87.46 \pm 1.12$ | $89.86 \pm 0.42$ | $91.97 \pm 4.15$ | $92.40 \pm 2.55$ |
| ABMIL | Baseline | $88.49 \pm 1.22$ | $87.97 \pm 1.04$ | $81.26 \pm 2.39$ | $79.36 \pm 1.72$ |
| | + Ours | $89.12 \pm 0.88$ | $90.58 \pm 0.68$ | $87.05 \pm 3.05$ | $82.52 \pm 2.33$ |
| DTFD | Baseline | $87.50 \pm 1.12$ | $88.52 \pm 0.68$ | $81.69 \pm 1.73$ | $80.77 \pm 0.62$ |
| | + Ours | $89.46 \pm 2.35$ | $90.12 \pm 1.97$ | $86.27 \pm 3.48$ | $84.29 \pm 1.63$ |
| TransMIL | Baseline | $88.44 \pm 2.27$ | $87.48 \pm 2.19$ | $79.90 \pm 2.66$ | $78.26 \pm 1.48$ |
| | + Ours | $88.77 \pm 2.81$ | $90.12 \pm 1.53$ | $78.11 \pm 2.84$ | $80.79 \pm 1.70$ |
| ABMIL-Aug | Baseline | $87.71 \pm 0.92$ | $88.52 \pm 0.31$ | $81.35 \pm 1.19$ | $79.70 \pm 0.22$ |
| | + Ours | $90.54 \pm 0.88$ | $91.14 \pm 1.18$ | $86.53 \pm 0.69$ | $82.69 \pm 0.66$ |
| DTFD-Aug | Baseline | $87.64 \pm 0.44$ | $88.27 \pm 0.09$ | $82.21 \pm 0.98$ | $80.37 \pm 0.17$ |
| | + Ours | $85.95 \pm 2.44$ | $88.63 \pm 1.58$ | $87.56 \pm 2.59$ | $86.04 \pm 2.07$ |
| TransMIL-Aug | Baseline | $87.29 \pm 0.56$ | $86.61 \pm 0.47$ | $81.52 \pm 0.15$ | $80.81 \pm 0.83$ |
| | + Ours | $88.77 \pm 2.97$ | $91.20 \pm 0.39$ | $79.71 \pm 4.38$ | $83.05 \pm 1.09$ |

**Runtime benchmark and scalability analysis.** We conduct a runtime benchmark to characterize how per-iteration training time (forward + backward + optimiser step) scales with input resolution, and how the addition of *Big-Layer* stages affects scalability. Experiments use ResNet-18 and ResNet-50 (Appendix A.2). We report wall times measured on an NVIDIA RTX 3090 (24 GB) for ResNet-18 and an NVIDIA A100 (80 GB) for ResNet-50. We sweep input side length from 4096 to 32768 pixels. We show the ResNet-18 results in Figure 2, and we present the corresponding ResNet-50 results in the Appendix in Figure 3. For each configuration, we report the mean per-iteration wall time with shaded bands indicating $\pm 1$ standard deviation computed from 10 repeats after 5 warmup iterations.

Two main trends emerge from the results. First, the baseline configuration—where the entire model resides on the GPU—exhibits near-linear scaling of iteration time with image area. Second, progressively enabling more *Big-Layer* stages increases the maximum feasible input size but also leads to higher per-iteration cost. The increase in cost is modest (sub-linear) for configurations using only two Big-Layers and becomes substantially steeper for configurations involving one or more Big-Layer stages. For ResNet-18, the fitted power-law exponents (see Appendix A.2) range from $b \approx 0.83$ to $1.48$, indicating sub-linear to super-linear growth. For ResNet-50, the exponents exhibit greater variability across configurations, spanning from $b \approx 0.83$ to $2.24$. We provide detailed fit statistics and coefficient of determination in Appendix A.2.

## 5.4 DISCUSSION AND FUTURE WORK

While we do not evaluate ViT architectures or fine-tune foundation encoders in this study, our partition-and-offload approach can support memory-efficient attention and encoder fine-tuning (see

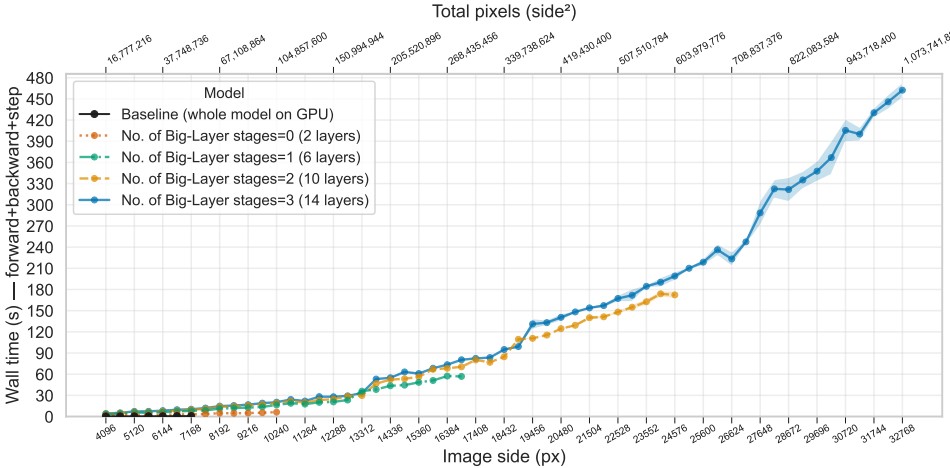

Figure 2: Incremental-stage benchmark for ResNet18: mean wall time per training iteration (forward + backward + optimizer step) versus input side length. Shaded bands denote $\pm 1$ standard deviation across repeated iterations. The top axis reports image area (side$^2$). Each curve corresponds to a model configuration where the number in brackets indicates the number of Big-Layers the model uses. Benchmarks were run on an NVIDIA RTX 3090 (24 GB). We report the fitted power-law exponents $b$ and doubling factors in Appendix A.2.

section 3.3), and we leave systematic evaluation of ViTs and encoder fine-tuning to future work. Because our method enables training the encoder, it also readily accommodates domain-generalisation techniques that require encoder training — for example, Representation Self-Challenging (Huang et al., 2020), Learning to Diversify (Wang et al., 2021), Correlated Style Uncertainty (Zhang et al., 2024), and nucleus-focused training (Tomar et al., 2024) — as well as augmentation strategies developed for H&E images (Marini et al., 2023; Shen et al., 2022).

While we focus on MIL for histopathology in this work, our method is broadly applicable. It allows training on WSIs without dividing them into patches, which can be helpful in applications where global context is essential.

The memory constraints we address are not unique to histopathology. For instance, remote sensing images can have a large image size, which poses challenges in tasks like segmentation Huang et al. (2018) and object detection Li et al. (2022). Our approach can be useful in these and other applications as well.

## 6 CONCLUSION

We present a practical method that leverages CPU RAM as auxiliary memory to overcome GPU memory limits for very large inputs such as whole-slide images. By integrating Big-Layers into standard MIL pipelines we enable end-to-end training of the encoder on gigapixel-scale inputs that previously required freezing or aggressive tiling.

Empirically, we demonstrate that end-to-end training with Big-Layers substantially improves predictive performance on large-scale histopathology benchmarks. On PANDA we report QWK gains of roughly 7–15 percentage points relative to frozen-encoder baselines; on CAMELYON our method improves external-set accuracy by up to $\sim$6 percentage points, indicating improved cross-site generalisability.

The runtime experiments clarify the principal trade-off: Big-Layers enable processing of inputs that exceed GPU memory at the cost of increased per-iteration runtime. Baseline (all-on-GPU) configurations exhibit near-linear scaling with image area, whereas using Big-Layers increases per-iteration cost.

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

# A  APPENDIX

## A.1  AREA UNDER CURVE FOR CAMELYON 17/16

We report AUCs for the CAMELYON datasets in Table 3.

Table 3: AUC on the CAMELYON17 dataset's test set and CAMELYON16 whole dataset. We train three models per method and report the mean and standard deviation of AUC. We report AUCs using 2 different inference setups: 1) Using all foreground tiles of the WSIs 2) Using only up to 2048 radomly selected tiles per WSI. For the 2048-tiles per bag setup, we test each model 10 times and use the mean AUC of the 10 runs as the final AUC for that model. "-Aug" indicates the use of augmentation.

| Method | | CAM17-All | CAM17-2048 | CAM16-All | CAM16-2048 |
|---|---|---|---|---|---|
| H-optimus-1 | ABMIL | $95.64 \pm 0.52$ | $96.00 \pm 0.41$ | $97.98 \pm 0.22$ | $95.76 \pm 0.42$ |
| | DTFD | $95.10 \pm 1.30$ | $95.76 \pm 0.69$ | $97.93 \pm 0.15$ | $95.76 \pm 0.29$ |
| | TransMIL | $95.71 \pm 1.91$ | $96.67 \pm 0.36$ | $98.22 \pm 0.32$ | $95.89 \pm 0.58$ |
| Prov-GigaPath | ABMIL | $95.81 \pm 0.69$ | $95.43 \pm 0.24$ | $97.66 \pm 0.14$ | $94.29 \pm 0.23$ |
| | DTFD | $95.61 \pm 0.18$ | $95.26 \pm 0.21$ | $97.68 \pm 0.36$ | $93.61 \pm 0.32$ |
| | TransMIL | $95.90 \pm 1.39$ | $96.29 \pm 0.52$ | $97.77 \pm 0.27$ | $95.20 \pm 0.75$ |
| UNI2-h | ABMIL | $96.36 \pm 0.48$ | $96.70 \pm 0.21$ | $98.60 \pm 0.04$ | $96.16 \pm 0.18$ |
| | DTFD | $95.95 \pm 0.94$ | $96.11 \pm 0.69$ | $98.53 \pm 0.27$ | $96.06 \pm 0.43$ |
| | TransMIL | $93.50 \pm 1.83$ | $96.09 \pm 1.45$ | $98.50 \pm 0.13$ | $96.35 \pm 0.46$ |
| ABMIL | Baseline | $89.72 \pm 1.94$ | $88.10 \pm 1.94$ | $81.96 \pm 1.60$ | $79.46 \pm 1.71$ |
| | + Ours | $94.76 \pm 0.60$ | $92.36 \pm 1.06$ | $87.80 \pm 2.66$ | $80.68 \pm 3.54$ |
| DTFD | Baseline | $92.10 \pm 0.28$ | $89.93 \pm 0.36$ | $83.02 \pm 1.16$ | $79.06 \pm 1.02$ |
| | + Ours | $94.62 \pm 1.36$ | $93.81 \pm 0.45$ | $89.72 \pm 3.61$ | $87.11 \pm 0.96$ |
| TransMIL | Baseline | $89.70 \pm 1.82$ | $88.82 \pm 1.68$ | $76.59 \pm 2.49$ | $74.12 \pm 2.90$ |
| | + Ours | $93.65 \pm 1.73$ | $92.92 \pm 1.63$ | $79.62 \pm 4.20$ | $79.34 \pm 3.50$ |
| ABMIL-Aug | Baseline | $91.77 \pm 0.37$ | $90.41 \pm 0.56$ | $80.82 \pm 0.48$ | $78.00 \pm 0.40$ |
| | + Ours | $92.20 \pm 1.00$ | $91.06 \pm 0.43$ | $85.96 \pm 3.03$ | $81.66 \pm 1.17$ |
| DTFD-Aug | Baseline | $92.38 \pm 0.36$ | $90.37 \pm 0.26$ | $85.48 \pm 1.22$ | $82.29 \pm 0.87$ |
| | + Ours | $94.75 \pm 1.18$ | $92.83 \pm 1.17$ | $87.56 \pm 2.59$ | $89.30 \pm 2.13$ |
| TransMIL-Aug | Baseline | $90.96 \pm 0.50$ | $89.10 \pm 0.53$ | $81.45 \pm 2.53$ | $80.09 \pm 2.41$ |
| | + Ours | $94.06 \pm 1.25$ | $93.31 \pm 0.86$ | $85.01 \pm 4.38$ | $83.79 \pm 1.86$ |

## A.2  RUNTIME SCALING ANALYSIS

To analyse how per-iteration runtime $t$ scales with input resolution, we fit the model

$$t(A) = a\,A^b + c,$$

where $A$ denotes image area (pixels), $a \geq 0$ is a scale factor, $b \geq 0$ is the power-law exponent and $c \geq 0$ is an additive overhead to capture constant per-iteration costs.

For each model configuration we report:

- the fitted parameters $a, b, c$;
- the coefficient of determination $R^2$;
- the *doubling factor* computed at the median tested area,

$$F_{\mathrm{dbl}} = \frac{a(2A_{\mathrm{med}})^b + c}{a(A_{\mathrm{med}})^b + c},$$

which reports how many times slower an iteration becomes when the image area doubles;
- the number of measured points $n$ used in the fit.

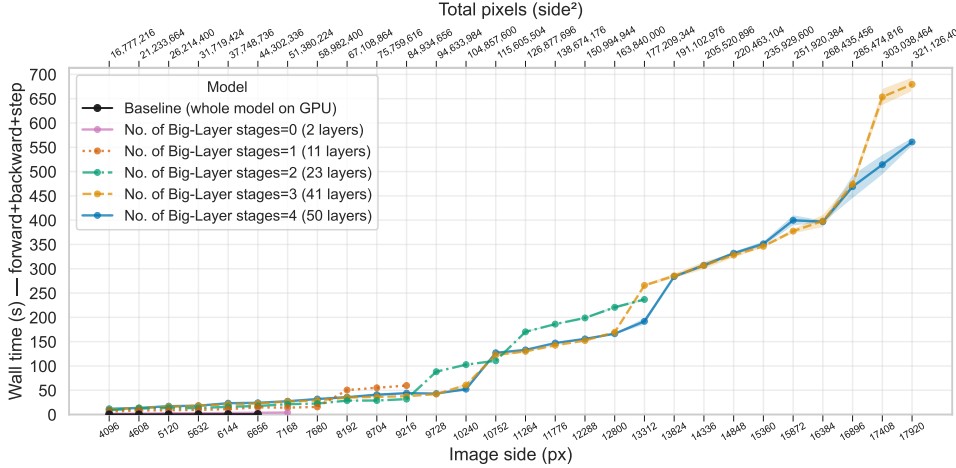

Figure 3: Incremental-stage benchmark for ResNet50: mean wall time per training iteration (forward + backward + optimizer step) versus input side length. Shaded bands denote $\pm 1$ standard deviation across repeated iterations. The top axis reports image area (side$^2$). Each curve corresponds to a model configuration where the number in brackets indicates the number of Big-Layers the model uses. Benchmarks were run on an NVIDIA A100 (80 GB).

Table 4: Power-law-plus-offset fit results for per-iteration wall time, $t(A) = a\,A^b + c$. Columns list the model configuration, number of measured points $n$, fitted parameters $a, b$ and $c$, coefficient of determination $R^2$ (original scale), and doubling factor $F_{\mathrm{dbl}}$ (time multiplier when area doubles at the median tested area). See Section A.2 for benchmarking details.

| Model | n | a | b | c | R$^2$ | F$_{\mathrm{dbl}}$ |
|---|---|---|---|---|---|---|
| **ResNet-18** | | | | | | |
| Baseline (whole model on GPU) | 7 | $7.43 \times 10^{-9}$ | 1.03 | $1.69 \times 10^{-2}$ | 0.99 | 2.00 |
| No. of Big-Layer stages = 0 (2 Big-Layers) | 13 | $1.23 \times 10^{-6}$ | 0.83 | $1.54 \times 10^{-4}$ | 0.98 | 1.78 |
| No. of Big-Layer stages = 1 (6 Big-Layers) | 26 | $1.67 \times 10^{-11}$ | 1.48 | 2.99 | 0.98 | 2.48 |
| No. of Big-Layer stages = 2 (10 Big-Layers) | 41 | $2.40 \times 10^{-9}$ | 1.24 | $3.21 \times 10^{-15}$ | 0.99 | 2.36 |
| No. of Big-Layer stages = 3 (14 Big-Layers) | 57 | $7.96 \times 10^{-11}$ | 1.41 | 5.37 | 0.99 | 2.57 |
| **ResNet-50** | | | | | | |
| Baseline (whole model on GPU) | 6 | $2.24 \times 10^{-8}$ | 1.00 | $7.86 \times 10^{-3}$ | 0.99 | 1.98 |
| No. of Big-Layer stages = 0 (2 Big-Layers) | 7 | $1.80 \times 10^{-6}$ | 0.83 | $3.21 \times 10^{-4}$ | 0.97 | 1.77 |
| No. of Big-Layer stages = 1 (11 Big-Layers) | 11 | $2.64 \times 10^{-19}$ | 2.57 | 2.63 | 0.93 | 5.04 |
| No. of Big-Layer stages = 2 (23 Big-Layers) | 19 | $5.24 \times 10^{-14}$ | 1.90 | $7.07 \times 10^{-4}$ | 0.97 | 3.74 |
| No. of Big-Layer stages = 3 (41 Big-Layers) | 29 | $5.00 \times 10^{-15}$ | 2.01 | 9.30 | 0.97 | 3.77 |
| No. of Big-Layer stages = 4 (50 Big-Layers) | 28 | $3.13 \times 10^{-12}$ | 1.68 | $4.14 \times 10^{-14}$ | 0.99 | 3.20 |

## A.3 Convolution pass using our method

**Note:** patches include halo (overlap) rows to guarantee correct convolution outputs at patch boundaries; during the backward pass overlapping gradient contributions to $\nabla X$ are accumulated (reduced) when stitching. Gradient accumulation for $\nabla W$ is performed incrementally on the chosen accumulation device (GPU or CPU) to bound peak memory usage.

---

**Algorithm 3** Memory-Efficient Convolution (Forward Pass)

---

1: **procedure** BIGCONV2DFORWARD($X, W$, stride, padding, dilation, max_N)
2:     **Determine** maximal patch height from max_N and model dimensions.
3:     **for** each batch slice of $X$ **do**
4:         **for** each (height) patch of $X$ (include halo rows as needed for kernel support) **do**
5:             Transfer the patch to GPU (non-blocking).
6:             Compute local convolution:

$$Y_{\text{patch}} \leftarrow \text{Conv2D}(X_{\text{patch}}, W, \text{stride}, \text{padding}, \text{dilation})$$

7:             Transfer $Y_{\text{patch}}$ back to CPU and place it into its position in $Y$.
8:         **end for**
9:     **end for**
10:     **return** Concatenated output $Y$ (stitching overlaps / halos if present).
11: **end procedure**

---

**Algorithm 4** Memory-Efficient Convolution (Backward Pass)

---

1: **procedure** BIGCONV2DBACKWARD($X, W, \nabla Y$, stride, padding, dilation, max_N)
2:     **Determine** maximal patch height from max_N and model dimensions.
3:     Initialize $\nabla W \leftarrow 0$ (accumulator on device) and $\nabla X \leftarrow 0$ (host or preallocated buffer).
4:     **for** each batch slice of $X$ **do**
5:         **for** each (height) patch of $X$ (use same partitioning/halo policy as forward) **do**
6:             Extract corresponding slice $\nabla Y_{\text{patch}}$ (output gradient) for this patch.
7:             Transfer $X_{\text{patch}}$ and $\nabla Y_{\text{patch}}$ to GPU.
8:             Compute partial gradients on GPU:

$$\Delta W \leftarrow \text{Conv2D\_weight\_grad}(X_{\text{patch}}, \nabla Y_{\text{patch}})$$

$$\nabla X_{\text{patch}} \leftarrow \text{Conv2D\_input\_grad}(\nabla Y_{\text{patch}}, W)$$

9:             Transfer $\Delta W$ to the accumulator location and update $\nabla W$ (reduce/accumulate).
10:             Transfer $\nabla X_{\text{patch}}$ to host and add into the corresponding slice of $\nabla X$ (accumulate where patches overlap).
11:         **end for**
12:     **end for**
13:     **return** $\nabla X, \nabla W$ (and $\nabla b$ if bias is present).
14: **end procedure**

---

