# OpenReview forum: "Big-Layers: Enabling end-to-end training"
_ICLR.cc/2026/Conference — ICLR 2026 Conference Withdrawn Submission_

### Official Review · Reviewer_zcrD · 2025-10-22

**Soundness:** 3
**Presentation:** 3
**Contribution:** 1
**Rating:** 2
**Confidence:** 4

**Summary:**

This paper proposes an approach for enabling end-to-end training on Whole Slide Images by offloading intermediate comptuations to CPU. Their experiments, performed on the Resnet-18 backbone, reveals consistent improvement over a frozen backbone. They propose layer-specific approaches for offsetting GPU memory requirements. Unfortunately, their experiments are solely performed on the CNN backbone, while current SOTA is dominated by vision transformers. For instance, their approach with ABMIL achives a quadratic weighted kappa of 84.60, while using the frozen UNI encoder achieves a performance of ~94 with ABMIL. The generalization of their approach to the self-attention mechanism of vision transformers is not discussed nor intuitive from their proposed algorithm, substantially limiting the utility of this approach.

**Strengths:**

- The ability to train models end-to-end is of substantial interest to the computational pathology community.
- The model is evaluated on three MIL methods, indicating that performance is not specific to a specific mechanism.
- The model is evaluated on a sufficient number of tasks.

**Weaknesses:**

- Performance is only shown on the Resnet-18 encoder, which is substantially behind the state of the art vision encoders (e.g UNI-2, Virchow-2, Conch 1.5). Consequently, even the improved end-to-end model lags behind frozen vision encoders by a wide margin.
- The way to apply this method to self-attention is not clear from the text, further limiting the relevance of these results to current state of the art.
- There are no benchmarks against other end-to-end approaches in CPath. There is consequently no evidence that this method provides benefit over existing approaches.
- There is substantial room in the paper for additional details. An improved description of the method should be provided.

As it stands, the work is not framed in terms of the state-of-the-art, and I believe there is not sufficient time to include such state-of-the-art. I am recommending reject as a result. However, I would be willing to update my rating if my comments are sufficiently addressed.

**Questions:**

- The code should be provided to help us further evaluate its quality and usability.
- The benchmark protocol does not need to be in bullet points. The extra space should be used to describe the method in greater detail.

---

> ### Author Response · Authors · 2025-11-21
> **Response to Reviewer zcrD**
>
> We thank the reviewer for the detailed and constructive comments.
>
> ### Weaknesses
>
> **1. Concern: performance below frozen SOTA encoders**
> We now include three foundation model baselines (Tables 1,2). These encoders perform strongly in-domain but still show domain-gap vulnerabilities between PANDA and TCGA-PRAD datasets.
>
> **2. Method applicability to ViTs**
> See Common Response §3. We now explain how self-attention can be implemented with our method (naïve blockwise or FlashAttention-style) in section 3.3. A full set of experimentation with our approach and ViTs is left for future work.
>
> **3. Benchmarks against alternative end-to-end methods.**
> We appreciate the reviewer’s request for direct benchmarks. We note that several prior approaches are conceptually different or complementary rather than direct drop-in alternatives:
>
> - a unified-memory implementation is a natural point of comparison, but, as discussed in Section 2 (lines 75-84), stable unified-memory APIs suitable for reproducible experiments are not available in current TensorFlow~2 or PyTorch releases, making a fair comparison difficult to implement reproducibly;
> - streaming/tiled convolution schemes and optimiser/state offload (ZeRO-style) address related resource constraints, but the StreamingCNN approach cannot train layers that require global statistics (like BatchNorm) while ZeR0-offload cannot handle inputs for which the forward pass activations do not fit in the GPU VRAM. Therefore, they are not direct alternatives to our approach. However, they can be combined with our method in many architectures (for example, streaming can be used for consecutive convolutional layers inside our partition-and-offload pipeline).
> - partial-backward methods trade gradient fidelity for lower memory and thus solve a different point in the accuracy–resource Pareto frontier.
>
> We have expanded the empirical scope in other ways by adding comparisons to three public foundation encoders (Table 1,2) and a ResNet-50 runtime scaling study (please see section 5.3).
>
> **3. On “no evidence of benefit over existing approaches.”**
> We respectfully disagree with the reviewer’s characterisation. As detailed in our Related Work, existing approaches do *not* support full end-to-end training of all layers for very large inputs: streaming/tiled convolution cannot train layers that require global statistics (e.g., BatchNorm), partial-backward methods compute approximate gradients for only a subset of patches, and optimiser/state-offload methods target model-state memory rather than activation memory. Our method’s benefit is therefore *capability-level*: it enables full backpropagation through **all** layers—including normalisation layers—on inputs that otherwise do not fit in GPU memory. Unified-memory training would be a direct comparison, but stable TensorFlow~2 and PyTorch APIs for controlling unified memory are currently unavailable. Whether this added capability yields performance improvements on any given public dataset depends on dataset-specific factors. For instance, our newly included results in Tables 1 and 2 show that models trained only on PANDA using our method outperform multiple foundation models on TCGA-PRAD. In contrast, the same is not the case between CAMELYON17 and CAMELYON16. From a technical standpoint, the benefit of our method is clear and distinct from prior work.
>
> **4. Request for more detail on the method**
> We thank the reviewer for their interest in the method. To improve clarity, pseudocode for the memory-efficient convolution has been added to the appendix. We think this addition provides the necessary clarifications, but please let us know if additional details are needed.
>
> ---
>
> ### Questions
>
> **1. Availability of code**
> All our cleaned-up code will be made publicly available and open-sourced upon acceptance of the paper.
>
> **2. Formatting of the paper**
> Thank you for the feedback. We have now revised the benchmark section accordingly.

---

> > ### Comment · Reviewer_zcrD · 2025-11-24
> >
> > I thank the authors for their detailed response, which has partially addressed my concerns. However, I maintain that evaluating performance with ViTs is essential to properly situate this work, as ViTs represent the current state-of-the-art in computational pathology. Given that all three reviewers have requested for evaluation on ViTs, I hope the authors will recognize its importance for properly contextualizing the work.
> >
> > Furthermore, benchmarking against existing end-to-end approaches is required to provide necessary context regarding both performance and runtime efficiency. While I acknowledge that the proposed approach may theoretically support a broader set of layers, empirical evidence demonstrating this benefit is necessary to substantiate the method's utility for the community.

---

### Official Review · Reviewer_VRyv · 2025-10-30

**Soundness:** 2
**Presentation:** 3
**Contribution:** 1
**Rating:** 2
**Confidence:** 3

**Summary:**

The paper proposes a new approach for making predictions on gigapixel histopathology images. Instead of the classical pipeline: (1) tiling slides into patches, (2) embedding patches with pretrained encoders, and (3) aggregating for slide-level prediction; the authors address GPU memory constraints by offloading intermediate activations to CPU RAM when GPU memory is insufficient. This enables joint training of the patch encoder and the slide-level predictor. Concretely, they apply this only to the first layers of a ResNet-18, stopping once sufficient downsampling allows the remainder to fit on the GPU.

**Strengths:**

The paper tackles an important limitation of Multiple Instance Learning (MIL) methods in computational pathology that tile whole-slide images into instances. It might be desirable to perform end-to-end training, which usually is not done with current workflows.

**Weaknesses:**

1. The paper is not well motivated within the existing literature. The community’s reliance on pretrained encoders stems not only from memory limits but also from limited labeled data, where pretraining provides substantial gains. The work cites a 2023 pipeline for MIL training, while the GPU memory bottleneck citations are from 2020–2021; the related work feels uneven and insufficiently up to date.
2. The primary encoder, ResNet-18, is dated for this domain.
3. The main comparison pits a fully trained ResNet-18 against a frozen ImageNet-pretrained ResNet-18. A fairer baseline would include contemporary frozen encoders pretrained on histopathology patches (e.g., UNI, Virchow, H-Optimus) to assess whether encoder finetuning is truly necessary and beneficial.
4. The approach is stated to be incompatible with ViTs, which are common encoders in this area. It would be important to evaluate compatibility and performance with larger CNNs (e.g., ResNet-50) and, if possible, discuss extensions to transformer-based models.
5. The runtime comparison is debatable. Since the baseline reportedly does not benefit from data augmentation; the time to be compared with should take into account the fact that the embeddings can be stored.

**Questions:**

1. Is the method practical with larger backbones such as ResNet-50?
2. How does the approach compare against strong frozen pretrained encoders (UNI, Virchow-2, H-Optimus)?
3. Given that the current methods not only respond to memory limits but also to limited labeled data, could the authors specify and demonstrate the use case where their method improves on current SOTA ?

---

> ### Author Response · Authors · 2025-11-21
> **Response to Reviewer VRyv**
>
> We thank the reviewer for the detailed and constructive comments.
>
> ### Weaknesses
>
> **1. “Paper not well motivated / related work uneven.”**
> We thank the reviewer for this comment. We have now added a dedicated discussion of recent foundation-model encoders (UNI, Virchow, H-Optimus, GigaPath, etc.) and their robustness limitations in the revised Introduction.
>
> We would like to clarify that the citations from 2020–2021 (Pinckaers 2020; Skrede 2020; Chen 2021) remain the *most relevant prior work* on **end-to-end WSI training under GPU memory constraints**. Although the field has progressed rapidly in terms of encoder architecture (2022–2025 foundation models), the underlying **memory bottleneck is still unchanged**: even the newest foundation encoders cannot be trained end-to-end on gigapixel WSIs and therefore continue to rely on frozen patch-level inference.
>
> To our knowledge, no recent (post-2021) work proposes a general solution that enables full forward *and backward* passes on entire WSIs, or allows training layers whose activations do not fit in GPU memory. The papers we cite therefore remain the most technically relevant references for methods attempting to overcome this bottleneck.
>
> Our method directly extends this line of work by providing a solution for training all the layers of a ConvNet end-to-end.
>
> **2. “ResNet-18 is dated.”**
> We chose ResNet-18 to demonstrate the practicality, correctness and reproducibility of our approach across many runs and datasets within our available compute budget. ResNet-18 allows extensive hyperparameter sweeps and repeat trials to establish stable performance gains. To show scalability beyond ResNet-18, we have now added **ResNet-50 runtime scaling curves (A100)** in the section 5.3 (with the table in the Appendix). Conceptually, our method is backbone-agnostic; empirical ViT / large-backbone fine-tuning is important future work (see point 4 below).
>
> **3. “Baseline should include contemporary frozen encoders (UNI, Virchow, H-Optimus).”**
> We have now added experiments comparing to three public foundation encoders (H-Optimus-1, UNI2-h, Prov-GigaPath). These appear in the revised tables (PANDA / TCGA-PRAD: Table 1; CAMELYON: Table 2) with AUCs in the Appendix. These comparisons place our results in context: foundation encoders often excel in-domain, but their cross-cohort performance varies by dataset and encoder (see Common Response §1 and the revised Results discussion).
>
> **4. “Approach incompatible with ViTs / need larger CNNs.”**
> We clarify in the Limitations section (Section 3.3). Please see also Common Response §3.
>
> **5. “Runtime comparison is debatable; precomputed embeddings should be considered.”**
> We have now revised **Training epoch comparison** paragraph in the Speed section. Please see also Common Response §2.
>
> ---
>
> ### Questions
>
> **Q1. Is the method practical with larger backbones such as ResNet-50?**
> Yes, the method is conceptually backbone-agnostic and scales to larger CNNs, subject to available device memory and wall-clock cost. To demonstrate scaling behaviour, we included ResNet-50 runtime curves measured on an A100 in section 5.3 (plot in Appendix). Training many ResNet-50 models end-to-end requires larger GPU pools than we currently have; the runtime study shows how costs grow and can help practitioners plan resources.
>
> **Q2. How does the approach compare against strong frozen pretrained encoders (UNI, Virchow-2, H-Optimus)?**
> Please see Common Response §1.
>
> **Q3 and W1**
> We agree that end-to-end encoder training might not improve performance in the case of limited labelled data. However, enabling end-to-end encoder training in settings with extreme input sizes such as WSIs removes a barrier which may be important in  practical use cases where strong domain-specific features matter and the available foundation models do not capture the relevant domain distribution as well as in settings where adaptations of foundation models or other encoders is warranted because of robustness or other requirements. Our results show that models trained only on PANDA using our method outperform multiple foundation models on TCGA-PRAD, demonstrating a case where end-to-end supervised adaptation (even with limited labels) yields better out-of-domain performance than current SOTA frozen encoders. As emphasised in the Common Response §1, the aim of this work is **not** to claim universal state-of-the-art on every benchmark. Rather, we provide a method that enables supervised encoder training on target data, giving practitioners an alternative to frozen foundation features.
>
> Furthermore, the size of histopathology datasets is only expected to grow over time, with larger datasets such as PANDA becoming available. This will make our method increasingly relevant for end-to-end training on high-resolution WSIs.

---

> > ### Comment · Reviewer_VRyv · 2025-11-27
> > **Unclear Benefit**
> >
> > First, I would like to thank the authors for addressing my concerns. I acknowledge that they performed the additional experiments I requested and reported the results with care and sincerity.
> >
> > However, the new results demonstrate that existing solutions (frozen embeddings obtained from a current foundation model) provide substantially better performance than the approach proposed in this paper, while also being orders of magnitude more efficient. As such, although the technical idea is clever and original, I do not currently see a clear practical or methodological benefit.
> >
> > It is possible that an application scenario exists in which the proposed technique would fill an important gap and solve a real-world problem. However, for an ICLR publication, I would expect such a use case and its benefit over existing methods to be clearly demonstrated. For this reason, I believe that publication of this article at this time would be premature. I will adjust the individual ratings accordingly, but not the final recommendation.

---

### Official Review · Reviewer_zGhW · 2025-10-30

**Soundness:** 2
**Presentation:** 3
**Contribution:** 3
**Rating:** 2
**Confidence:** 4

**Summary:**

The authors propose an approach to train the patch encoders on gigapixel WSI data with label supervision. This requires backpropagating gradients from the WSI label to the patch encoder layers, for which they propose dynamically offloading intermediate activations of a layer to CPU RAM, allowing the layer to process inputs that do not fit in the GPU memory. The authors show improved performance on PANDA, TCGA-PRAD and CAMELYON in terms of quadratic weighted kappa metric, compared to an ImageNet pre-trained encoder baseline, at a cost in training time.

**Strengths:**

Significance: The method allows backpropagating supervised signal back to the patch encoders of WSI for potentially significant gains in performance on WSI classification tasks. This type of strategy is interesting and could potentially help design very strong encoders in the future.

Clarity: The paper is generally clear.

**Weaknesses:**

The main weaknesses currently have to do with the evaluation of the proposed method. Reinforcing the evaluation either during the rebuttal phase or after resubmission would change my assessment of the paper.

1. The single baseline is a ResNet-18 encoder pre-trained on ImageNet, which artificially inflates the gain in performance from the proposed approach. The field has largely moved to task-specific encoders trained with self-supervised learning (DINO, MoCov3, etc.), or to encoders based on foundation models (UNI, Virchow, GigaPath, etc.) trained on large numbers of WSIs from various histopathology datasets. These encoders should be used as baselines instead.

2. I am not convinced by the fairness of the comparison in terms of runtime. The authors report 45 minutes for one epoch for the baseline on PANDA vs. 200 minutes for the proposed approach. But my understanding is that this is when computing patch "embeddings on the fly on GPU" also for the baseline, whereas patch embeddings are usually computed offline and saved once and for all in the standard approach (as no backpropagation is required). Hence training the MIL stage from pre-computed patch embeddings for the baseline could be significantly faster.

3. A nice addition would be to also demonstrate the method on a ViT encoder and on a ResNet-50, to get a sense of the scalability and flexibility of the approach. Experiments are so far limited to ResNet-18.

**Questions:**

The major questions that could be answered to change my initial rating have to do with the weaknesses listed above.

Additional questions:
- Could the authors also report AUCs when relevant for direct comparison with the literature?

---

> ### Author Response · Authors · 2025-11-21
> **Response to Reviewer zGhW**
>
> We thank the reviewer for the thoughtful and constructive feedback.
>
> ### Weaknesses
>
> **1. Stronger baselines / foundation encoders**
> Please see Common Response §1. We now include results for three stronger histopathology-specific foundation encoders.
>
> **2. Runtime fairness for precomputed embeddings**
> Please see Common Response §2. We now explicitly report the epoch time when using precomputed-embeddings (~20 s/epoch).
>
> **3. Larger backbones / ViTs**
> Please see Common Response §3. ResNet-50 runtime results are included in section 5.3 with a plot in the Appendix. ViT applicability is now discussed explicitly in the section 3.3.
>
> ### Questions
>
> **1. Request for AUC metrics**
> We thank the reviewer for this comment, we have included AUC results in Table 3 in the appendix.

---

> > ### Comment · Reviewer_zGhW · 2025-11-24
> >
> > I thank the authors for their detailed response to the reviewers' comments.
> >
> > Many thanks in particular for including foundation models and AUCs. These comparisons however tend to highlight current limitations of the method.
> >
> > More importantly ultimately, it is still difficult to evaluate the empirical benefit of the proposed method, because there is no direct quantification (if I am not mistaken) of the direct gain going from a strong pre-trained patch encoder (by SSL or from a foundation model) to the same patch encoder + supervised training. This experimental setup could also be more advantageous for the authors than the current comparison between patch encoder from a foundation model vs. patch encoder randomly initialized + supervised training.

---

### Author Response · Authors · 2025-11-21
**Common Response to All Reviewers**

We thank all reviewers for their constructive and detailed comments. The major common concerns are addressed below.

### 1. Stronger baselines (foundation models)
All reviewers requested comparisons to modern foundation encoders. We have now added experiments with three foundation models: H-Optimus-1, UNI2-h, and Prov-GigaPath. Results are included in the revised tables:

- PANDA / TCGA-PRAD results: **Table 1**
- CAMELYON results: **Table 2**
- AUC metrics: **Appendix Table 3**

We observe that the in-domain performance of foundation models on PANDA is state-of-the-art, as expected. However, when applied cross-domain to TCGA-PRAD, the performance drops substantially and is similar to or lower than our end-to-end trained models. This inadequate robustness and domain generalisation is consistent with recent studies on foundation models, as now mentioned in section 1 with references [1-5] below. In contrast, foundation encoders generalise well from CAMELYON17 to CAMELYON16, likely reflecting a smaller domain gap.

We would like to emphasise that our goal is **not** to claim universal SOTA but to **enable practitioners to train encoders end-to-end**, providing an alternative when frozen features underperform.

Moreover, **training encoders enables the use of domain-generalisation methods**, which we now discuss in the revised section 5.4. Incorporating such methods can further improve out-of-domain robustness and may allow end-to-end trained encoders to outperform foundation models on a wider range of datasets. However, systematically exploring combinations of domain-generalisation algorithms with encoder training across many datasets requires a **community-wide effort**, and we view this work as enabling that future exploration rather than attempting to benchmark all possible combinations.

---

### 2. Runtime comparisons of the precomputed-embeddings baseline and scalability to large models
Reviewers zGhW and VRyv noted that our original runtime comparison did not explicitly include the standard “precompute embeddings once” baseline. The original speed comparison intended to evaluate our layer implementations relative to PyTorch’s native GPU-resident implementations (which do not offload to the CPU), but we agree that reporting all relevant runtime numbers provides a clearer picture. We have now created a **Training epoch comparison** paragraph reporting on three setups tested on PANDA with an NVIDIA RTX 3090 GPU card:
- **Precomputed-embeddings baseline:** ~20 s/epoch (train MIL head only)
- **Baseline, on-the-fly embeddings (no encoder backprop):** ~45 min/epoch
- **Our end-to-end method (with encoder backprop):** ~200 min/epoch

Although our method supports training larger encoders, doing so requires proportionally larger GPU memory and more computation time. Currently, we do not have sufficient numbers of large-memory GPUs to train many high-capacity models such as ResNet-50 at scale. Nevertheless, to demonstrate scalability, we performed a **runtime scaling study for ResNet-50 on an NVIDIA A100** GPU card and reported the results in section 5.3 (with plot in the Appendix).

---

### 3. ViTs, larger backbones, and applicability
All reviewers asked about applying our method to ViTs and larger encoders.

We now explicitly clarify in section 3.3 that our method *can* support self-attention and also outline **two practical implementation paths**:

1. **Naïve self-attention:** treat linear projections as 1×1 convolutions (applied on 1D input) and apply the same sub-tensor offloading strategy as for CNN layers.
2. **IO-aware tiled attention (FlashAttention):** load Q/K/V blocks (sub-tensors) to GPU, compute block-level attention, and update outputs incrementally (also, this is compatible with the partition-and-offload design in our proposed method).

Implementing and benchmarking ViTs with our method remains an important direction, as doing so would also enable **fine-tuning existing foundation models** using our approach. Exploring this direction requires substantial additional engineering and experimentation, and we therefore leave a full empirical study to future work.

[1] de Jong et al. "Current pathology foundation models are unrobust to medical center differences." arXiv preprint arXiv:2501.18055 (2025).

[2] Kömen, Jonah, et al. "Do histopathological foundation models eliminate batch effects? A comparative study." arXiv preprint arXiv:2411.05489 (2024).

[3] Kömen, Jonah, et al. "Towards robust foundation models for digital pathology." arXiv preprint arXiv:2507.17845 (2025).

[4] Gustafsson, Fredrik K. et al. "Evaluating computational pathology foundation models for prostate cancer grading under distribution shifts." arXiv preprint arXiv:2410.06723 (2024).

[5] Filiot, Alexandre, et al. "Distilling foundation models for robust and efficient models in digital pathology." International Conference on Medical Image Computing and Computer-Assisted Intervention. Cham: Springer Nature Switzerland, 2025.

---

### Note · Authors · 2025-11-28

**Comment:**

We thank the reviewers for their feedback. While the goal of this work was to enable full end-to-end supervised training of gigapixel WSIs rather than to achieve SOTA performance on every benchmark, several reviews emphasised SOTA-level comparisons across datasets. Some public benchmarks may already be close to saturation, but our method did outperform multiple foundation encoders in certain out-of-domain settings (in particular, models trained on PANDA evaluated on TCGA-PRAD). Regardless, we appreciate the feedback and plan to explore the directions suggested by the reviewers.

**Withdrawal Confirmation:**

I have read and agree with the venue's withdrawal policy on behalf of myself and my co-authors.